# Identifying Depression-Related Behavior on Facebook—An Experimental Study

**Zoltán Kmetty** [1,*] and **Károly Bozsonyi** [2]

1   Computational Social Science—Research Center for Educational and Network Studies, Sociology Department and Centre for Social Sciences, Faculty of Social Sciences, Eötvös Loránd University, 1117 Budapest, Hungary

2   Institute of Social and Communication Sciences, Faculty of Social Sciences, Károli Gáspár University of the Reformed Church in Hungary, 1091 Budapest, Hungary; bozsonyi.karoly.bk@gmail.com

*   Correspondence: kmetty.zoltan@tatk.elte.hu

**Abstract:** Depression is one of the major mental health problems in the world and the leading cause of disability worldwide. As people leave more and more digital traces in the online world, it becomes possible to detect depression-related behavior based on people's online activities. We use a novel Facebook study to identify possible non-textual elements of depression-related behavior in a social media environment. This study focuses on the relationship between depression and the volume and composition of Facebook friendship networks and the volume and temporal variability of Facebook activities. We also tried to establish a link between depression and the interest categories of the participants. The significant predictors were partly different for cognitive-affective depression and somatic depression. Earlier studies found that depressed people have a smaller online social network. We found the same pattern in the case of cognitive-affective depression. We also found that they posted less in others' timelines, but we did not find that they posted more in their own timeline. Our study was the first to use the Facebook ads interest data to predict depression. Those who were classified into the less interest category by Facebook had higher depression levels on both scales.

**Keywords:** depression; PHQ-9; Facebook; social media; digital data





## 1. Introduction

Depression is one of the major mental health problems globally and the leading cause of disability worldwide (Mathers and Loncar 2006). Based on Eurostat (https://ec.europa.eu/eurostat/web/productseurostatnews//DDN201703231?inheritRedirect=true&redirect=/eurostat/; accessed on 15 January 2022) data, 7 percent of the European people experience current depression symptoms. Among the European countries, Hungary (the focus country of this paper) had the highest rate of depressed people (10 percent). On top of that, major depression is the most common psychiatric disorder in people who commit suicide (Hawton et al. 2013). Untreated depression causes more health issues than treated depression (Wang 2004). Early detection of depression is crucial for rapid intervention, as it can reduce the long-lasting effects of this mental health problem (Le and Boyd 2006).

Parallel with the increased access to digital footprint data, more and more studies analyzed how depression and mental health problems could be monitored through online traces. Three primary digital traces were studied: search engines (McCarthy 2010; Hagihara et al. 2012; Kristoufek et al. 2016; Tran et al. 2017), forums (Al-Mosaiwi and Johnstone 2018; De Choudhury et al. 2016; De Choudhury and Kiciman 2017), and social media platforms. We will focus on the latter in this study.

One part of the study analyzes the content of social media posts and pictures but does not have any information about the users who post these contents (Chancellor et al. 2016; Moreno et al. 2016; Kmetty et al. 2017; Brown et al. 2018). The general findings of these studies are that people use depression and mental health-related words and hashtags in many different ways and content contain suicide and depression-related words and

hashtags only weakly connected to actual suicidal ideation and/or depression (Koltai et al. 2020; Spates et al. 2020; O'Dea et al. 2015).

The other part of the study tries to classify depressed and healthy people based on their social media activity. Our paper follows the same logic, so we present these studies in more detail. First, we start with those papers which tried to link social media activities with a broadly defined depression phenomenon. Then we would focus on a narrower view of depression through suicidal tendencies.

Uban et al. (2021) tested different classifiers on Reddit data. They built linguistic-based models on anorexia, self-harm, and depression. All three had different explanatory variables. They found that depression is the hardest to classify. Another significant result was that too old posts do not help classify mental health issues.

De Choudhury et al. (2013) acquired a group of Twitter users who report being diagnosed with clinical depression. Their social media postings over a year preceding the onset of depression measured behavioral attributes relating to social engagement, emotion, language and linguistic styles, ego network, and mentions of antidepressant medications. Based on these cues, the researchers built a statistical model that estimates the risk of depression before the reported onset. They found that social media contains proper signals for characterizing the onset of depression in individuals, as measured by decreased social activity, raised negative effects, highly clustered ego networks, heightened relational and medicinal concerns, and greater expression of religious involvement. Reece and Danforth (2017) analyzed Instagram photos and successfully classified people into healthy/depressed and healthy/non-diagnosed depressed groups. Images posted by depressed participants were grayer, darker, and bluer. Those who had depression received more comments and fewer likes on their posts.

Interestingly, people with depression posted more photos with faces but had a lower average face count per picture. Healthy people used fewer filters and brighter filters compared with depressed participants. Schwartz et al. (2014) used survey responses and status updates from 28,749 Facebook users to develop a regression model that predicts users' degree of depression based on their Facebook status updates. The model's user-level predictive accuracy was modest ($r = 0.386$), but significantly outperformed the baseline of average user sentiment ($r = 0.149$). They used this model to estimate user changes in depression across seasons and found, consistent with the literature, that users' degree of depression most often increases from summer to winter. Eichstaedt et al. (2018) downloaded the history of Facebook statuses posted by 683 patients visiting an urban emergency department. One hundred fourteen of whom had a diagnosis of depression. Mainly using the language preceding their first documentation of a diagnosis of depression, they could identify depressed patients with fair accuracy ($AUC = 0.69$), approximately matching the accuracy of screening surveys benchmarked against medical records. They found that predicting future depression status was possible as far as three months before its first documentation. They did not find a specific daily temporal posting pattern within those who had depression compared to the rest of the sample. Park et al. (2015), based on online Facebook logs of 212 young adults, showed that the size of the network ($r = -0.25$) and the frequency ($r = -0.24$) of interactions on social networks have close associations with depression. Depressed individuals have smaller involved networks regarding comments and likes. In contrast to the decreased interactions, depressed individuals increased the wall post rates ($r = 0.26$) and were active online during midday.

Cheng et al. (2017) collected survey and post data from Weibo users. They found that lower usage of verbs and frequent use of pronouns correlate with high suicide probability. Huang et al. (2017) also examined the Chinese social media accounts (Weibo) of 130 people who committed suicide between 2011 and 2016. They found an observable but slight change in content leading up to the time of death. People posted more suicide-related words and more posts with negative sentiment value before their death. They also found that before their suicide, people posted more than earlier. Braithwaite et al. (2016) analyzed the data of the Twitter feed of MTurk participants and were able to classify the risk of

suicide with 92 percent accuracy. They had two interesting results—lower frequency of religion-related words and achievement-related words correlated with higher suicide risk. Joshi and Patwardhan (2020) also analyzed Twitter data. They collected tweets from people at depression risk and compared them with "normal" users. They used machine learning models (word-embedding, fuzzy-clustering) to classify the users. Their best model achieved a 76 percent accuracy. Regarding posting time, they had different results than Park et al. (2015). Joshi and Patwardhan (2020) found that people at risk post more tweets in the evening.

The studies presented in the introduction illustrate that many studies have already looked at the relationship between social media activity and depression. A common feature of these studies is that they use textual elements to estimate exposure to depression almost without exception. These analyses based on textual data are highly country/culture/language-specific. The only exception is the Park et al. (2015) study, in which the authors analyzed the association of non-textual Facebook data with depression indicators. Our analysis uses similar logic, but we move to a more complex empirical dataset. In addition to the activity variables, we analyze data related to friends and acquaintances and the participant interest categories identified by Facebook. The Park et al. (2015) study included only university students, and the time window was six months. Our analysis considers a two-year time window, and our sample is more general, including not only university students. We expect a much more complex approach than previous research can reveal exciting and new findings on the relationship between social media activity and depression.

## 2. Materials and Methods

Most of the major social media platforms available in Western countries allow users to access their data through Data Download Packages (DDP). Our study used this option and asked 150 Hungarian Facebook users to export their own Facebook data and share it with us. The sample is a non-probability quota sample with an age quota. The original plan included at least 30 percent older than 30 years of people in the sample. In the final sample, 33 percent were older than 30 years. There was one substantial selection criterion in the study. The participants had to be regular Facebook users (use the platform at least weekly). A professional market research company did the fieldwork from April to September 2019. Study participants received an incentive of 10 Euros. The market research company used its existing participant pool for recruitment. In addition, they also recruited people through student cooperatives. Before starting the study, participants were asked to sign an informed consent form. Unfortunately, we do not have any information about those who rejected the participation. We know from other studies that the dropout rate could go up to 80 percent (Breuer et al. 2022).

The participants were asked to log into their Facebook account and then download their Facebook data-profile archive in JSON (JavaScript Object Notation) format. The data collected in this study covers a wide range of Facebook activities: posts, comments, likes, and reactions, liked pages, friends, profile information, and data about ads. The data covers the whole time of the participants' Facebook use. Preserving participants' privacy was a crucial issue in this data collection process. Raw data were anonymized right after the export to ensure anonymity. In addition to sharing their Facebook data, participants were asked to fill out an online questionnaire. This questionnaire included various political questions, media use, self-representation, spare-time activities, and music preferences. Above that, we asked the participants to fill out a modified version of the Patient Health Questionnaire (PHQ-9). The same ID code was used for the online questionnaire and stored Facebook data to link these two data sources.

As mentioned above, we use a slightly modified version of the PHQ-9 module to identify the symptoms of depression among the study participants. This relatively short tool provides a reliable measurement of depression (Kroenke et al. 2001; Löwe et al. 2004). It was used in similar studies earlier see: (De Choudhury et al. 2013). The nine items of PHQ correspond to the nine DSM-IV diagnostic criteria for depression (Boothroyd et al.

2019). We modified the PHQ-9 module in two ways. Initially, the PHQ-9 questions focused on symptoms of the last two weeks.

We wanted to measure a more extended period, so we asked about symptoms from the last year. The original PHQ-9 aims to help capture current depression. Our aim was different; we wanted to know if respondents had depression-related symptoms in the previous year. The second modification is related to the answer categories. The original PHQ-9 had four categories: Not at all, Several days, More than half the days, and Nearly every day. We changed this 4-levels scale to a 7-levels scale (1: Never, 7: Very often) because we wanted an identical scale characteristic through the whole questionnaire. We created 7-levels scales across the entire questionnaire to create an easy-to-fill questionnaire. Answering different types of scales is a challenging cognitive process that could harm the quality of the survey. The study's objective was not the clinical diagnostic of depressed people but to provide a reliable measure of depression, so we think this alteration is acceptable.

We calculated the severity of depression in our sample based on the PHQ-9 standards (see Figure 1). 3.5 percent of the respondents had severe symptoms, 13.2 percent moderately severe, and 35 percent had mild symptoms. Based on this result, this measure has enough variability to use it as a dependent variable.

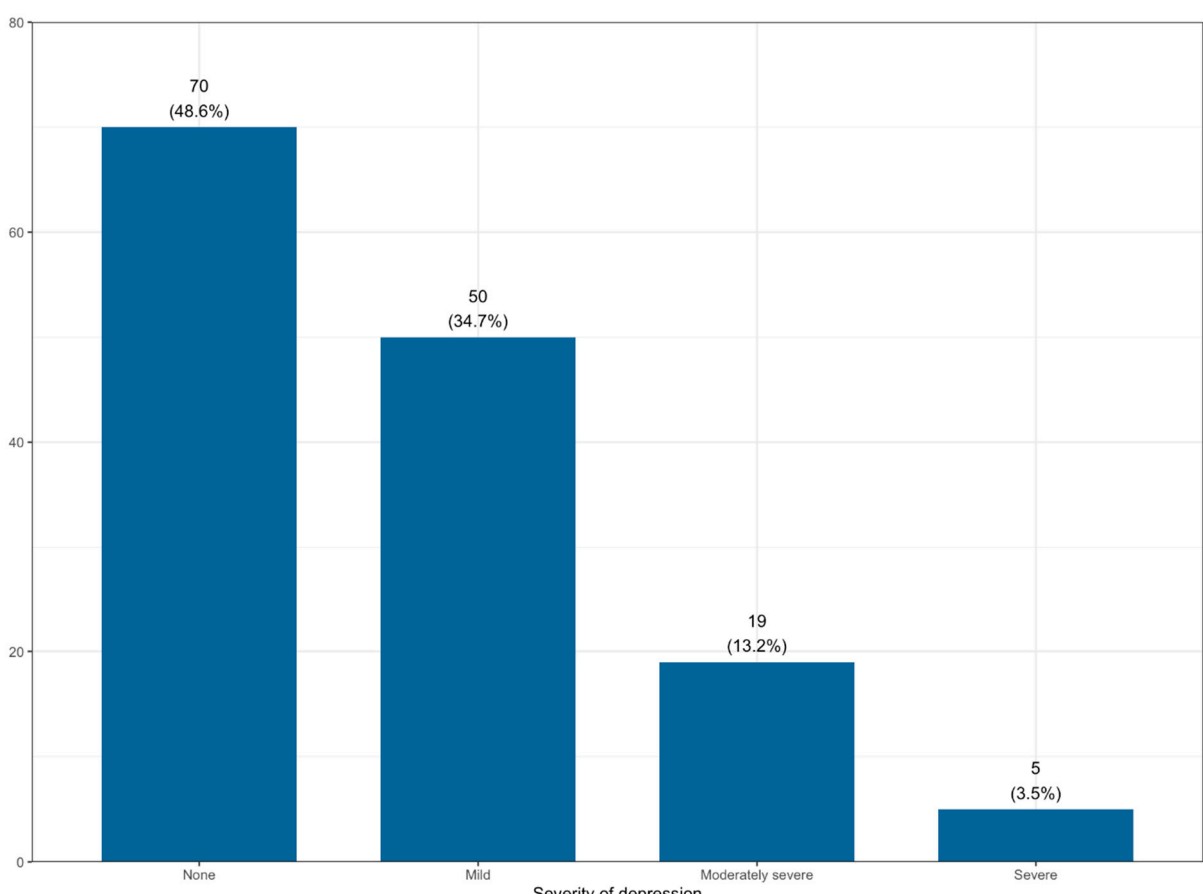

**Figure 1.** The severity of depression in the sample based on PHQ-9 standards. *n* = 144.

Factor analysis is usually used to calculate indicators based on the PHQ-9 question block. There is no consensus in the literature whether a single-factor or two-factor solution gives a better fit of PHQ-9 (Boothroyd et al. 2019). We could distinguish between mood-related depression indicators and somatic indicators in a two-factor solution. González-Blanch et al. (2018) found that both solutions give reliable results. Based on the initial analysis results, we fitted a confirmative factor model with two factors. Following the literature, we extracted a cognitive-affective depression factor and a somatic depression

factor (see Table 1). The RMSEA value of the initial factor model was 0.1, which value was far from good. Two variables from the somatic factor (Trouble concentrating on things, such as reading the newspaper or watching television AND moving or speaking so slowly that other people could have noticed) had a poor fit, so we decided to exclude them from the factor model. The final model fit statistics were good; the CFI (Comparative Fit Index) value was 0.99, the RMSEA (Root mean square error of approximation) was 0.036. The correlation between the factors was 0.72. The two depression factors were the dependent variables in the analysis.

**Table 1.** Confirmative factor model on the PHQ-9 module. $n = 144$.

| Factors | Variable | Factor Loadings |
|---|---|---|
| Cognitive-affective depression | Little interest or pleasure in doing things? | 0.55 |
| | Feeling down, depressed, or hopeless? | 0.90 |
| | Feeling bad about yourself—or that you are a failure or have let yourself or your family down? | 0.82 |
| | Thoughts that you would be better off dead, or thoughts of hurting yourself in some way? | 0.60 |
| Somatic depression | Trouble falling or staying asleep, or sleeping too much? | 0.74 |
| | Feeling tired or having little energy? | 0.80 |
| | Poor appetite or overeating? | 0.62 |
| CFI | 0.993 | |
| RMSEA | 0.036 | |

CFI = Comparative Fit Index; RMSEA = Root mean square error of approximation.

As described above, the survey was linked with the Facebook data of the participants. Several different indicators could be extracted from Facebook datasets. In this study, we focus on three areas: Friendship network, level of Facebook activities, and the interest categorization of the participants by Facebook. We tried to cover all aspects of these datasets to create the independent variables. The descriptive statistics of independent variables are reported in Table 2.

Friends: This data contains all Facebook friends of the participants, the timestamp for the start of the friendship, the hashed name of the friend, and the estimated gender of the friend based on the friend's first name. This dataset also contains information about rejected and pending friend requests and the number of removed friends.

We used the following independent variables in the analysis:

- number of actual Facebook friends
- the percentage of female friends
- gender homogeneity (absolute distance from a balanced (50 percent) gender distribution)
- rejected number of friend requests by the user (normalized with the number of friends)
- removed number of friends by user (normalized with the number of friends)

Based on Park et al. (2015), we expected smaller friend networks to correlate with higher depression levels. We did not have any literature-based hypothesis for the other variables, but we had expectations. We expected that gender homogeneity would correlate with higher depression levels. Social isolation means not just fewer friends but fewer friends from the opposite gender. In rejection and removal variables, we expected higher numbers to go along with more severe depression symptoms. Someone who has a terrible mood does not want to build large friend networks; they try to eliminate those not closely connected.

**Table 2.** Descriptive statistics of independent variables.

| Dimensions | Variables | *n* | Mean | Std.Dev | Std.Error | Median | Range | Skewness |
|---|---|---|---|---|---|---|---|---|
| Friends | Number of actual Facebook friends | 144 | 784.28 | 515.28 | 42.94 | 690.5 | (8–2840) | 1.33 |
| | Number of close friends | 144 | 2.71 | 7.1 | 0.59 | 0.61 | (0–45.67) | 4.55 |
| | Close friends variation | 116 | 1.63 | 1.76 | 0.16 | 1.06 | (0.06–14.5) | 4.48 |
| | Percentage of female friends | 144 | 0.54 | 0.12 | 0.01 | 0.55 | (0.12–0.85) | −0.37 |
| | Gender homogenity | 144 | 0.1 | 0.08 | 0.01 | 0.08 | (0–0.38) | 1.14 |
| | Rejected number of friend requests (normalized) | 144 | 0.2 | 0.81 | 0.07 | 0.05 | (0–8.75) | 9.24 |
| | Removed number of friends (normalized) | 144 | 1.3 | 11.04 | 0.92 | 0.05 | (0–132.12) | 11.81 |
| Reactions | Reaction on friends—total sum | 144 | 111.07 | 55.47 | 4.62 | 119.5 | (0–181) | −0.56 |
| | Reaction on friends—absolute change | 144 | 18.33 | 18.21 | 1.52 | 13 | (0–84) | 1.39 |
| | Reaction on friends—number of active days | 144 | 832.9 | 1657.69 | 138.14 | 350.5 | (0–13867) | 4.99 |
| | Reaction on pages—total sum | 144 | 43.38 | 49.92 | 4.16 | 19.5 | (0–172) | 1.13 |
| | Reaction on pages—absolute change | 144 | 16.89 | 21.66 | 1.8 | 9 | (0–123) | 2.25 |
| | Reaction on pages—number of active days | 144 | 152.81 | 416.73 | 34.73 | 24.5 | (0–4195) | 7.03 |
| Posting | Posts on own timeline —number of active days | 144 | 1.72 | 5.73 | 0.48 | 0 | (0–54) | 6.71 |
| | Posts on own timeline —total | 144 | 1.95 | 7.07 | 0.59 | 0 | (0–68) | 7.04 |
| | Posting on own timeline—absolute change | 144 | 1.31 | 2.87 | 0.24 | 0 | (0–19) | 4.26 |
| | Posts on other's timeline—number of active days | 144 | 18.77 | 26.82 | 2.24 | 8.5 | (0–158) | 2.75 |
| | Posts on other's timeline—total | 144 | 33.74 | 71.59 | 5.97 | 10 | (0–501) | 4.5 |
| | Posting on other's timeline—absolute change | 144 | 7.2 | 8.53 | 0.71 | 4 | (0–50) | 2.5 |
| Within day dynamic | Morning ratio | 143 | 0.17 | 0.11 | 0.01 | 0.15 | (0–0.64) | 1.66 |
| | Midday ratio | 143 | 0.2 | 0.08 | 0.01 | 0.19 | (0–0.54) | 0.98 |
| | Afternoon ratio | 143 | 0.21 | 0.08 | 0.01 | 0.21 | (0–0.67) | 1.39 |
| | Evening ratio | 144 | 0.31 | 0.13 | 0.01 | 0.31 | (0–1) | 1.31 |
| | Night ratio | 143 | 0.12 | 0.07 | 0.01 | 0.11 | (0–0.32) | 0.36 |
| Interest | Number of Interest categories | 144 | 25.6 | 7.34 | 0.61 | 28 | (0–33) | −1.95 |

The variables presented are based on the respondents' complete Facebook friends list. We also wanted to include indicators in the analysis built on closer Facebook friends. For this, we used the reaction dataset. We defined those as close friends where the number of reactions toward this Facebook friend was greater than 5 per month. We calculated the number of close friends based on this definition every month, between January 2018 and June 2019. Participants of the study started to use Facebook in different periods. If we go back longer, we would have missing data for some users who began to use Facebook later.

The first extracted indicator was the mean number of close Facebook friends; the second indicator was the variance of close FB friends between the months divided by the average number of close FB friends (variation coefficient). We wanted to measure the variability within the number of close friends with the second indicator. We expected that fewer close friends and a higher variation of friends might correlate with higher depression.

We set the minimum number of reactions at 5 per month for close friends as described above. We could not rely on any previous work, so we tried to apply two simple principles here: theoretical and empirical. A close friend means regular and reciprocal communication between two people. We could not measure reciprocity, but we could measure regularity. We would expect at least one reaction per week from a close friend. Of course, it could be more, but not less. This calculation set a minimum number for the limit, around 4–5. For the empirical principle, we considered the distribution of variables. We calculated how the average close friends' value and the proportion of no close friends are affected by the threshold choice. The results can be found in Table A1 in Appendix A. If we set a very low minimum response number, the meaning of "close friend" disappears. When we set the parameter to 1 reaction per month, there were 40 close friends of the sample members on average. We can set the threshold value as high as we want; however, the proportion of people with no friends increases significantly. When we set the threshold at 20, more than two-thirds of the sample has become "isolated." Setting the threshold at five is seemed to be an excellent middle ground, with an average close friend number of 2.8 and "only"

20 percent of the sample was isolated. We also tested different thresholds in the correlation analysis to control for robustness. We return to these in the Section 3.

The other set of independent variables was built on the Facebook activities of users. We used the reaction data and the posting data. The reaction data contains all the users' reactions, with a timestamp and the type of reaction and the target of the reaction (i.e., if it was a friend or a page). The posting dataset contains information on posting on someone else's timeline, posting in groups with a timestamp, and posting on their timeline and status updates.

For independent variables, we wanted to choose widespread activities among users. We divided the activities into four distinct categories: reaction on friends, reaction on pages, posts on their own timeline, and posts on others' timelines. Post on others' timelines includes the activities when someone wrote a message to a peer's Facebook wall. Most of these posts are birthday greetings. On own timeline, most activities are status updates and posts about current mood or specific events. We calculated indicators to measure the volume of these activities and the time variability of these activities. We expected that a lower activity level and more substantial time variability might sign depression.

We demonstrate the logic of our calculations using the posts on their own timeline data. We measured the posting activity volume with two variables. The first indicator was the sum of the posts on their own timeline between 1 January 2019 and 30 June 2019. As this variable was highly skewed, we wanted to calculate a more robust activity version. So, the second indicator was the number of days with at least one post between 1 January 2019 and 30 June 2019. We calculated the same variable with the latter logic for the 1 January 2018–30 June 2018 time period and calculated the absolute value of the difference between the active days of 2019 and 2018. In the initial analysis, we tested many operationalization options. It turned out that only the dynamic of variation matters, not its sign. That is why we used this absolute value.

We measured the activity change with this composite variable. We calculated the same indicators in the case of posts on others' timelines, reactions on friends, and reactions on pages.

Park et al. (2015) found that depressed people were more active in social media middays. To measure the daily temporal variation of the activities, we created five daily time periods:

- 6:00–9:59—morning
- 10:00–13:59—midday
- 14:00–17:59—afternoon
- 18:00–21:59—evening
- 22:00–5.59—night

We calculated the period activities' rate using all the activities data (reactions, posts) in 2019 (until June 30).

The last set of independent variables was based on the interest categorization of respondents. Facebook categorizes every user for advertising purposes. This categorization is an algorithmic classification of the users based on their likes, activities, used keywords, and their friends' preferences (DeVito 2017). The algorithm is a black box, so we can only observe the categorization result. There is no timestamp in this data set, just the category names per user. Facebook links every user with several categories; the average category number per user is above 100. There are general categories (vegan food) and more specific categories (Vegan burger). In this study, we use the general categories. It is straightforward to distinguish between general and specific categories. General categories start with a small letter, and specific categories start with a capital letter in the Facebook database. In our sample, we found 678 general categories.

First, the authors of this paper manually coded these items into 33 high-level categories and 138 sub-categories (for the distribution of these categories, see the online appendix). First, we tested clustering methods to find the patterns in the data. We could find our high-level categories based on the patterns and expert choices. We manually checked all interest

variables for the sub-level categorization and decided if they were close enough to form a sub-level category. We have to highlight that this categorization is valid for this specific Hungarian sample. We are sure that it is possible to create a more general categorization, but a more general sample is needed for more respondents. The first indicator we extracted from this dataset was the number of interest categories by the user. We expected that a lower number of interest categories might correlate with a higher depression level.

For a more detailed analysis, we used the sub-level categories. It is pretty easy to overfit a model; therefore, we followed a complex data processing design. We run a random forest analysis on the sub-categories (Breiman 2001; Sun et al. 2017; Cacheda et al. 2019). There was a statistical dependency between the number of interest category variables and the dichotomous sub-interest variables. To handle this possible confounding factor, we decided to use the residuals of the fitted regression models as dependent variables in the random forest analysis. We fitted separate models for the two depression indicators.

We split the sample into a train and a test set in the first step of the random forest analysis. Two-thirds of the participants were moved to the train set (96 people) and the rest to the control set. In the sampling process, we kept the proportion of gender. We maximized in five how many interest variables could enter into the model. We excluded variables from the model, those interest variables that had a lower correlation with the given depression scale than 0.1 (absolute value). We omit those variables where the incidence rate was over 90%. We grew 1000 trees per sample. We ranked all the variables based on their influence value in the model. We used the node purity influence statistic for the ranking. We repeated the process (starting from sampling) 1000 times to achieve a robust result. At the end of the modeling, we had average rank importance for all the interest variables.

We used two demographic variables for control purposes; the gender of the respondents (75 percent female) and a categorized age variable based on the survey answers. We coded one of those under 30 years (68 percent) and two of those under 30 and above. There was a significant correlation (Anova statistic, $p < 0.002$) between cognitive-affective depression and age. Those in the younger age cohort had a higher value on the cognitive-affective depression scale. We found the same relationship between the somatic depression scale and age (Anova statistic, $p < 0.001$). The ANOVA calculation also showed a significant correlation between somatic depression and gender ($p < 0.01$). Female participants had a higher value on this depression scale.

In the paper's analysis part, we first present the raw correlation of selected independent variables with the depression measures. Then we fit regression models with the most promising variables, and if it is possible, we extend these models with interaction effects of the selected indicators with age and gender. We present B (unstandardized) and Beta (standardized) coefficients in the regression tables. We performed a random forest analysis in the ads interest analysis to determine which interest categories correlate with depression level (see above). We used the R program to fit our models and plot the interaction figures.

Four respondents had not answered all the PHQ-9 questions. We omitted them from this study. We also omitted those 2 participants who started to use Facebook in 2018, as their data was too short for the dynamical analysis of activity. We omitted them because we needed a full 2018 year's data to calculate the statistics, and they had only partial data from that year. The final sample size was 144.

## 3. Results

### 3.1. Friendship Network and Depression

First, we analyzed the relationship between the depression measures and the volume and composition of the friendship network of the respondents.

First, we calculated the bivariate Pearson correlation between friendship network and depression indicators. We only highlighted those coefficients in Figure 2, where the significance level was below 0.1.

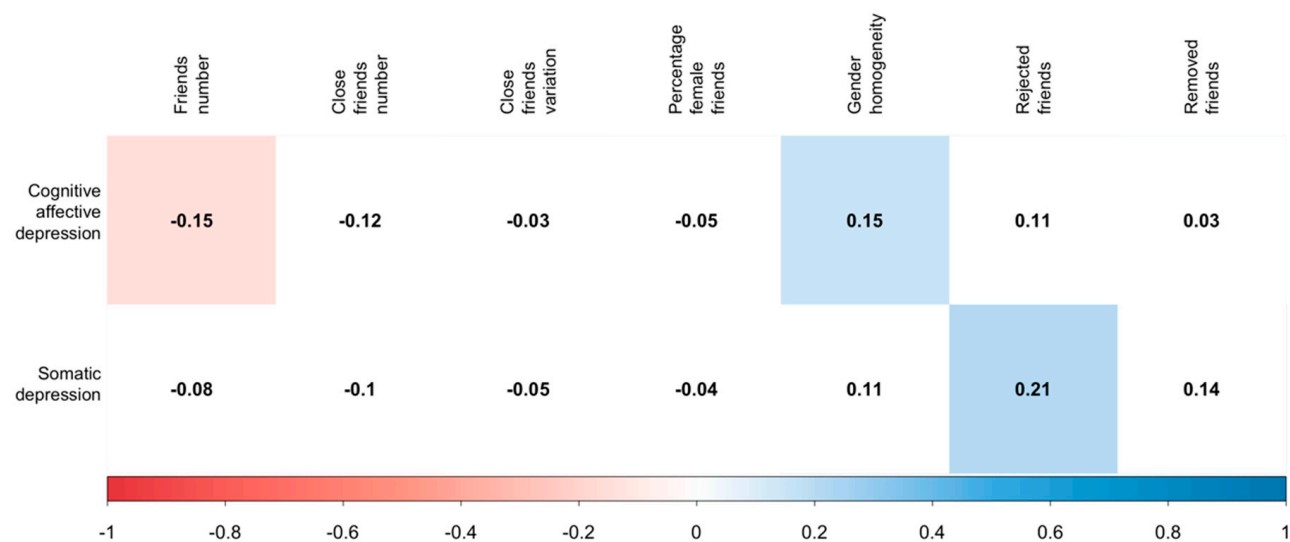

**Figure 2.** Pearson correlation between friendship network indicators and depression indicators.

Based on the bivariate statistics, there was a negative correlation between the **number of friends** and the cognitive depression variable and a positive correlation between **gender homogeneity** and cognitive depression. So those who had cognitive-affective depression symptoms had a smaller and more gender homogenous FB network.

The close friend's variables were not significant, such as the female friend's percentage ratio and the number of removed friends. The correlation coefficients were relatively modest; the highest value was 0.21. The small effect size means that we could only find a weak correlation between the variables.

We ran a regression model to test whether the results remain after controlling demographic variables (see Table 3). After controlling with gender and age, the effects were still significant, and the significance level of the network variables went below 0.01. Based on the beta coefficient, the explanatory power of the two network indicators was equal—it was around 0.2 (absolute value).

**Table 3.** Linear Regression model. Dependent variable: cognitive depression.

| Variables | B | Std.Error | Beta | Sig | B | Std.Error | Beta | Sig |
|---|---|---|---|---|---|---|---|---|
| (Intercept) | 0.8 | 0.47 | | 0.04 | 0.18 | 0.58 | | 0.68 |
| Number of actual Facebook friends | −0.04 * | 0.02 | −0.23 | 0.01 | −0.05 | 0.02 | −0.24 | 0.00 |
| Gender homogeneity | 2.8 | 1.00 | 0.23 | 0.01 | 8.7 | 3.10 | 0.70 | 0.00 |
| Gender | 0.11 | 0.17 | 0.05 | 0.51 | 0.18 | 0.17 | 0.07 | 0.36 |
| Age | −0.78 | 0.18 | −0.39 | 0.00 | −0.33 | 0.28 | −0.16 | 0.22 |
| Gender homogeneity * Age | | | | | −3.9 | 2.00 | −0.6 | 0.05 |
| Adjusted $R^2$ | 0.15 | | | | 0.17 | | | |

* We divided the friend number by 100, to help the interpretation of the regression effects. Without this the B coefficients of this variable would be close to zero.

We tested the interaction effects of the demographic variables and the two network variables. We found a significant interaction term between age and **gender homogeneity** (see Table 3). Using marginal models, we plotted the difference between the age categories (see Figure 3). Gender homogeneity only affected the younger age cohort; in the case of older respondents, there was no correlation between gender homogeneity and cognitive-affective depression.

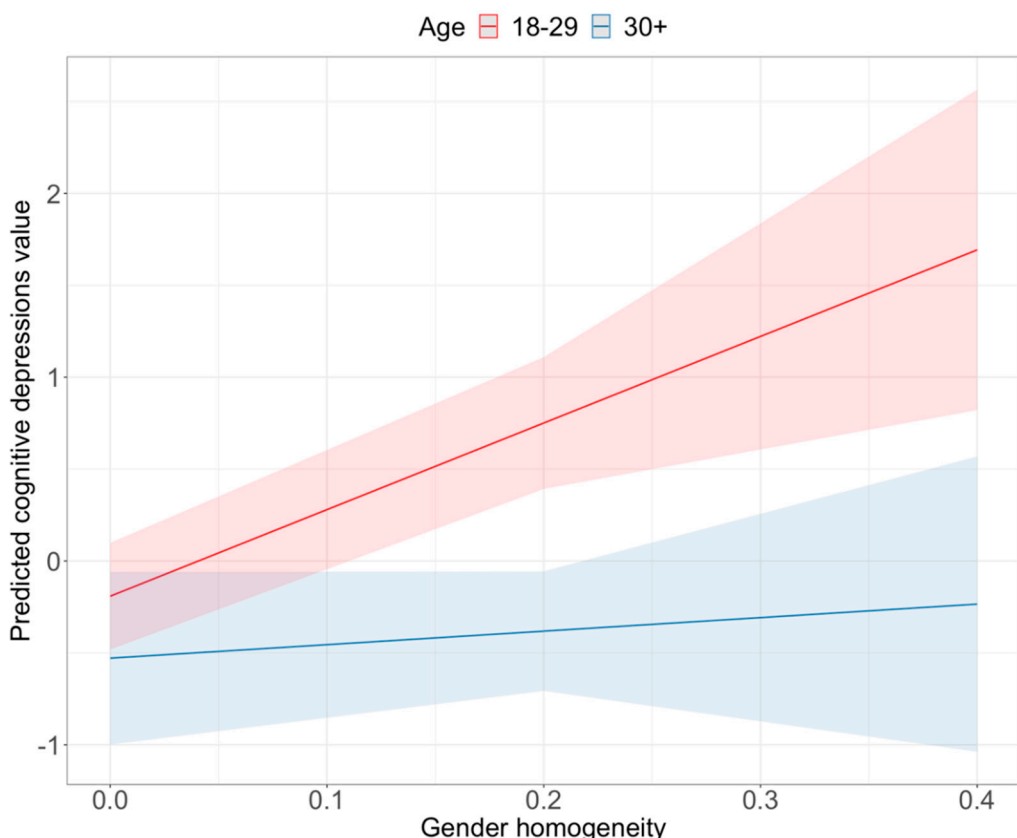

**Figure 3.** Interaction effect of gender homogeneity and age on cognitive depression (marginal models).

We fitted a linear regression model to test the relationship between **friend rejection** and somatic depression (see Table 4). After controlling the demographic variables, friend rejection was still significant, with a 0.24 beta coefficient. More friend rejection went along with higher scores on somatic depression. We did not find significant interaction effects here.

**Table 4.** Linear Regression model. Dependent variable: somatic depression.

| Variables | B | Std.Error | Beta | Sig |
|:---:|:---:|:---:|:---:|:---:|
| (Intercept) | −0.16 | 0.41 | | 0.71 |
| Rejected number of friend requests (normalized) | 0.27 | 0.08 | 0.24 | 0.03 |
| Gender | 0.37 | 0.17 | 0.18 | 0.03 |
| Age | −0.41 | 0.16 | −0.21 | 0.01 |
| Adjusted R$^2$ | | 0.12 | | |

### 3.1.1. Activity

There is a wide range of activities a user can do when using Facebook. In this analysis, we built our analysis on posts and reactions.

We started the analysis by exploring the bivariate correlations between the activity variables and the depression indicators (see Figure 4).

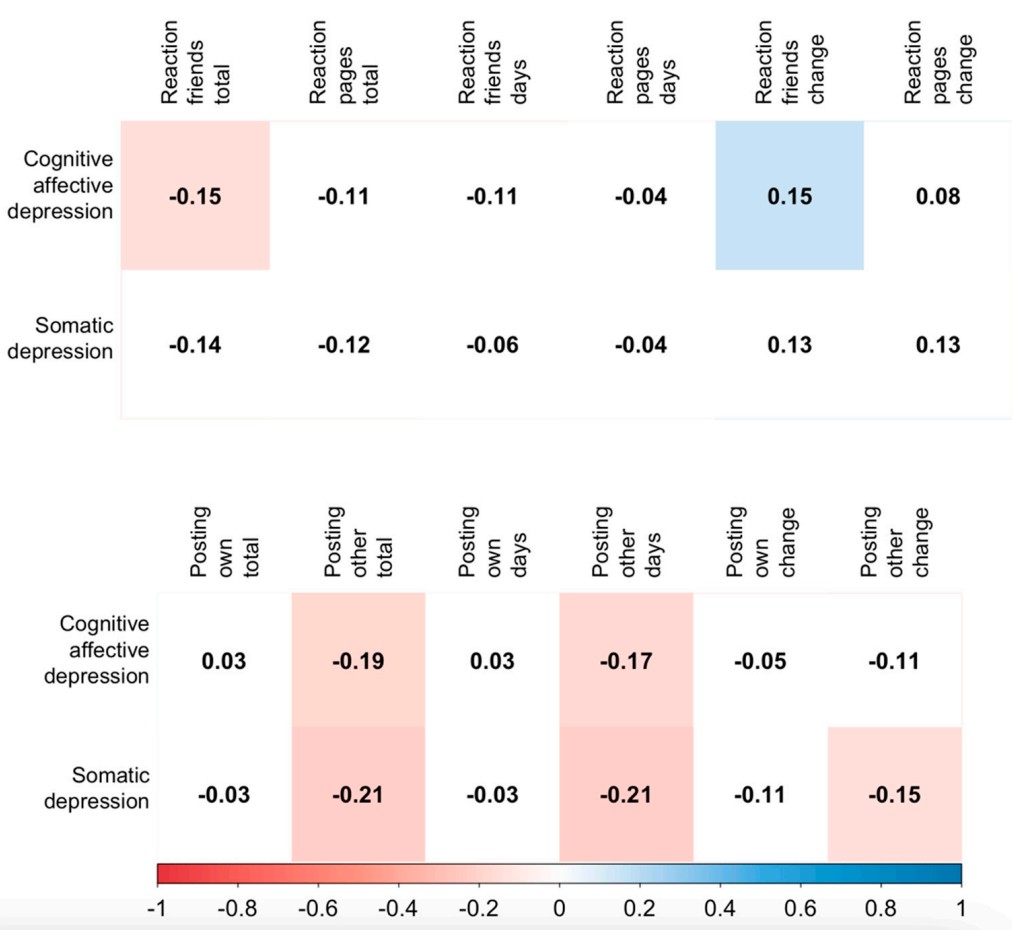

**Figure 4.** Pearson correlation between activity indicators and depression indicators.

Most of the observed correlations were not significant. The volume of reaction on pages and posts on their own timeline did not correlate with any depression indicators. In the case of **posts on others' timelines** and **reactions on friends**, we found a significant correlation with both depression types. Those who scored higher in depression scales had lower activity levels in these dimensions. We found a negative correlation between **absolute change of posting on others' timelines** and somatic depression. It seems that those who had higher somatic depression levels posted less on others' timelines, but the average volume of posting was more stable in their cases.

After controlling with demographic variables, all the significant correlations with cognitive-affective depression were diminished. For somatic depression, the volume of posts on other's timelines, the number of days on posting on other's timelines, and the absolute change of posting on other's timelines remained significant in a multivariate regression environment using a 90 percent confidence interval level. The two-volume indicator coefficients size were similar; we only present in this paper the regression model of the total number of posts on other's timeline indicators (see Table 5). The significant level of the posting activity variable was 0.08, and the beta coefficient was −0.15. The absolute change of posting on others' timelines was also significant with a negative coefficient. It seems that those who had higher somatic depression levels posted less on others' timelines, but the average volume of posting was more stable in their cases. We could not use both variables in one joint model because they had strong correlations (r = 0.51); that is why Table 5 consists of two models with partly different independent variables. We did not find any significant interaction effects with demographic variables here.

**Table 5.** Linear Regression model. Dependent variable: somatic depression.

| Variables | B | Std.Error | Beta | Sig | B | Std.Error | Beta | Sig |
|---|---|---|---|---|---|---|---|---|
| **(Intercept)** | −0.11 | 0.42 | | 0.79 | −0.02 | 0.58 | | 0.97 |
| Posts on other's timeline—total sum * | −0.19 | 0.11 | −0.15 | 0.08 | | | | |
| Posts on other's timeline *—abs. change | | | | | −1.5 | 0.85 | −0.14 | 0.08 |
| Gender | 0.32 | 0.18 | 0.16 | 0.06 | 0.35 | 0.17 | 0.17 | 0.05 |
| Age | −0.31 | 0.17 | −0.16 | 0.07 | −0.37 | 0.16 | −0.19 | 0.03 |
| Adjusted $R^2$ | | | 0.09 | | | | 0.09 | |

* We divided the Posts on other's timeline variables by 100, to help the interpretation of the regression effects. Without this the B coefficients of these variables would be close to zero.

We also tested the time period of activities. Only the **evening period** correlated with the cognitive-depression scale (see Figure 5).

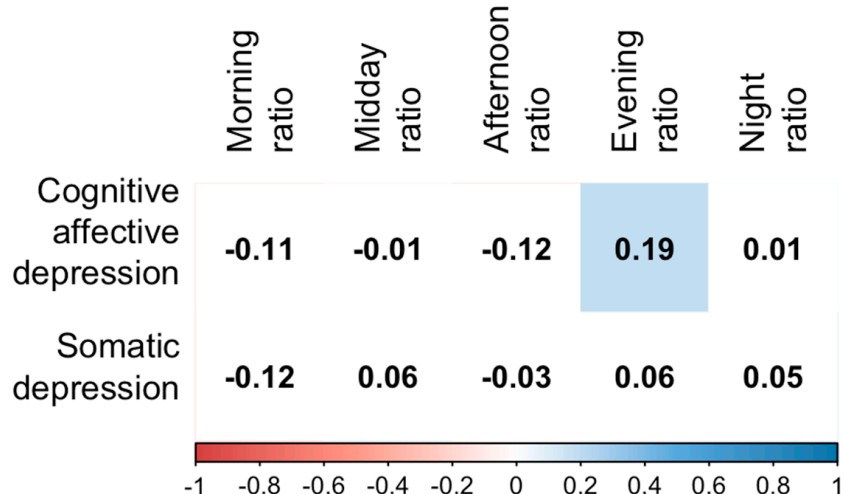

**Figure 5.** Pearson correlation between within day activity indicators and depression indicators.

The relationship between the evening ratio and the cognitive-affective depression relationship was still significant after controlling the demographic variables (see Table 6). Those who had higher cognitive-affective depression levels were more active in the evening period.

**Table 6.** Linear Regression model. Dependent variable: Cognitive-affective depression.

| Variables | B | Std.Error | Beta | Sig | B | Std.Error | Beta | Sig |
|---|---|---|---|---|---|---|---|---|
| (Intercept) | 0.08 | 0.47 | | 0.85 | −1.32 | 0.84 | | 0.11 |
| Evening ratio | 1.25 | 0.63 | 0.16 | 0.05 | 5.1 | 2.1 | 0.67 | 0.01 |
| Gender | 0.09 | 0.19 | 0.04 | 0.65 | 0.19 | 0.20 | 0.09 | 0.33 |
| Age | −0.46 | 0.17 | −0.22 | 0.01 | 0.35 | 0.44 | 0.17 | 0.43 |
| Evening ratio × Age | | | | | −2.55 | 1.26 | −0.66 | 0.05 |
| Adjusted $R^2$ | | | 0.07 | | | | 0.09 | |

The interaction of **the evening activity** ratio variable and age was significant ($p < 0.05$), with a negative coefficient (Figure 6). The evening activity only affected cognitive-affective depression levels in the younger age cohort.

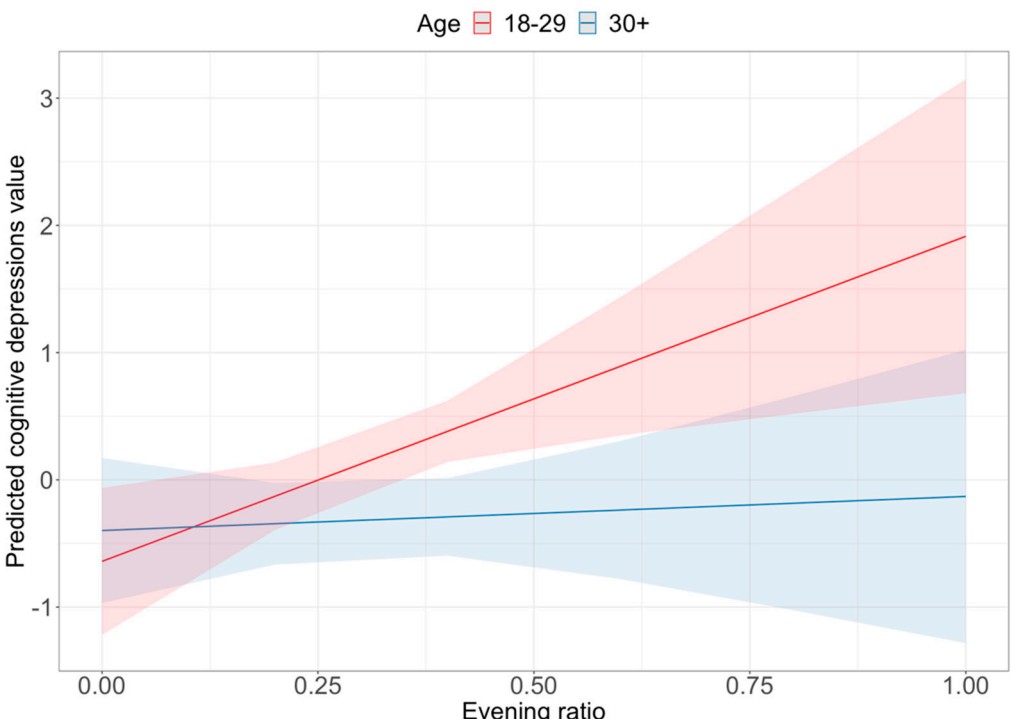

**Figure 6.** Interaction effect of evening activity ratio and age on cognitive-affective depression (marginal models).

3.1.2. Ads Interest

Facebook categorizes every user for advertising purposes. This categorization is available in the data archive. There is no clear definition of how the algorithm works; we can only observe the categorization results. There is no timestamp in this data set, just the category names per user.

The initial correlation analysis showed that those classified into **more high-level categories** had lower scores on depression scales. We tested these initial results with a regression model (see Table 7).

**Table 7.** Linear Regression models. Interest categories. Base model.

| Variables | Cognitive-Affective | | | | Somatic | | | |
|---|---|---|---|---|---|---|---|---|
| | B | Std.Error | Beta | Sig | B | Std.Error | Beta | Sig |
| (Intercept) | 1.04 | 0.48 | | 0.03 | 0.38 | 0.47 | | 0.41 |
| Number of Interest categories | −0.04 | 0.01 | −0.27 | 0.00 | −0.02 | 0.01 | −0.18 | 0.03 |
| Gender | 0.28 | 0.18 | 0.13 | 0.13 | 0.41 | 0.18 | 0.20 | 0.03 |
| Age | −0.47 | 0.17 | −0.23 | 0.01 | −0.38 | 0.16 | −0.20 | 0.02 |
| Adjusted R$^2$ | 0.12 | | | | 0.09 | | | |

The multivariate regression model (see Table 8) confirmed the result of the correlation analysis. Those linked to a wide variety of categories had lower cognitive-affective depression scores and a lower score on the somatic depression variable. We did not find significant interaction with any of the control variables.

For a more detailed analysis, we used the sub-level categories. As described in the Section 2, we applied a random forest analysis on the subcategories. We found three promising variables to model cognitive-affective depression. The first subcategory was about **beer and wine**. The second was about **cars and car racing**. And the third was about **cats and dogs**. The second one was not significant in the total sample regression model, so we focused on the other two variables in our analysis.

**Table 8.** Linear Regression models. Interest categories. Extended model.

| Variables | Cognitive-Affective | | | | Somatic | | | |
|---|---|---|---|---|---|---|---|---|
| | B | Std.Error | Beta | Sig | B | Std.Error | Beta | Sig |
| (Intercept) | 0.93 | 0.48 | | 0.04 | 0.09 | 0.45 | | 0.84 |
| Number of Interest categories | −0.03 | 0.01 | −0.23 | 0.01 | −0.05 | 0.01 | −0.37 | 0.00 |
| Gender | 0.27 | 0.18 | 0.12 | 0.14 | 0.63 | 0.18 | 0.3 | 0.00 |
| Age | −0.45 | 0.16 | −0.22 | 0.01 | −0.28 | 0.15 | −0.14 | 0.07 |
| Beer-wine interest category | −0.34 | 0.17 | −0.16 | 0.04 | | | | |
| Dogs-cats interest category | 0.31 | 0.16 | 0.16 | 0.05 | 0.35 | 0.15 | 0.19 | 0.02 |
| American sports | | | | | 0.5 | 0.16 | 0.27 | 0.00 |
| Adjusted $R^2$ | 0.15 | | | | 0.17 | | | |

The **beer-wine** sub-category had a negative coefficient; the **dogs-cats** sub-category had a positive one. Hence, the latter increased the cognitive-affective depression level, and the former decreased it.

We ran the same analysis with the somatic depression factor. We found two variables with high influence. One of them was the **dog-cat** sub-category, and the other was a **US sports sub-category** (such as American football or baseball). The **dog-cat** sub-category had a similar positive coefficient as in the cognitive-affective model. The other category was more interesting, perhaps a little bit contra-intuitive. The **US sports variable** had a positive coefficient, too, with a 0.27 beta. So those who were categorized here by Facebook scored higher on the somatic depression scale. This variable had a significant relationship with gender. Sixty percent of the male participants were categorized here and only 40 percent of the female (Chi2 $p < 0.05$). The beta coefficient of the gender variable increased after we added the US sports sub-category to the model. The interaction effect of the gender and the US sports category was significant ($p = 0.1$). The US sports sub-category only increased the female participants' somatic depression risk.

### 3.1.3. Joint Models

In the last part of the analysis, we created a joint model of friend's network, activity data, and ads interest data. We used those variables which were significant in the previous regression models.

In the joint model of cognitive-affective depression (see Table 9), only the **beer-wine interest category** significance level went above 0.1, the other indicators were significant at 0.05 level (except gender). The model explained more than 20 percent of the depression variable variance. All the model statistics (normality, heteroskedasticity) were acceptable; the multicollinearity was low (all the VIF values were below 2). Smaller and more gender homophilous networks, higher rate of evening activity, and lower number of ads interest categories were the significant predictors of cognitive-affective depression.

**Table 9.** Linear Regression model. Dependent variable: Cognitive-affective depression.

| Variables | B | Std.Error | Beta | Sig |
|---|---|---|---|---|
| (Intercept) | 0.72 | 0.50 | 0.00 | 0.15 |
| Gender | 0.26 | 0.19 | 0.12 | 0.17 |
| Age | −0.65 | 0.18 | −0.32 | 0.00 |
| Number of actual Facebook friends * | −0.04 | 0.02 | −0.19 | 0.03 |
| Gender homogenity | 2.05 | 1.01 | 0.17 | 0.04 |
| Evening ratio | 1.42 | 0.58 | 0.19 | 0.02 |
| Number of Interest categories | −0.04 | 0.01 | −0.28 | 0.00 |
| Beer-wine interest category | 0.19 | 0.16 | 0.10 | 0.22 |
| Dogs-cats interest category | 0.32 | 0.17 | 0.17 | 0.05 |
| Adjusted $R^2$ | | 0.205 | | |
| KS ** test of normality (residual) $p$-value | | 0.60 | | |
| Breusch-Pagan test $p$-value | | 0.59 | | |

* We divided the friend number by 100, to help the interpretation of the regression effects. Without this the B coefficients of this variable would be close to zero. ** Kolmogorov-Smirnov.

Somatic depression had different predictor variables (see Table 10). The **friend rejection variable** was significant in the joint model, but the posting activity significance level went above 0.1. The **lower number of ads interest** categories and the **dogs-cat interest** were both significant predictors. This result was very similar to the cognitive-depression model. The adjusted R2 was lower here—18.2 percent.

**Table 10.** Linear Regression model. Dependent variable: Somatic depression.

| Variables | B | Std.Error | Beta | Sig |
|---|---|---|---|---|
| (Intercept) | 0.13 | 0.44 | 0.00 | 0.77 |
| Gender | 0.43 | 0.17 | 0.21 | 0.01 |
| Age | −0.32 | 0.16 | −0.17 | 0.05 |
| Rejected number of friend requests (normalized) | 0.21 | 0.09 | 0.19 | 0.02 |
| Posting on others timeline—total sum * | −0.12 | 0.10 | −0.10 | 0.23 |
| Number of interest categories | −0.02 | 0.01 | −0.16 | 0.08 |
| Dogs-cats interest category | 0.32 | 0.15 | 0.18 | 0.03 |
| American sports | −0.25 | 0.16 | −0.14 | 0.11 |
| Adjusted $R^2$ | | 0.182 | | |
| KS test of normality (residual) *p*-value | | 0.50 | | |
| Breusch-Pagan test *p*-value | | 0.41 | | |

* We divided the Posts on other's timeline variables by 100, to help the interpretation of the regression effects. Without this the B coefficients of these variables would be close to zero.

## 4. Discussion

People post/share/create billions of content on social media every day, providing valuable data for researchers. We need mixed (social media + survey) data sources to understand the full spectrum of users' behavior and online space choices. We conducted a novel parallel data collection method, rarely used before, combining a face-to-face survey with personal FB data archives. Based on this novel data source, we attempted to identify the footprints of depressed behavior in an online social media environment.

We used three types of Facebook data: friends, activities, and ads interests. We found significant predictors of depression in all of the datasets. The predictors were partly different for cognitive-affective depression and somatic depression.

Earlier studies found (Park et al. 2015; De Choudhury et al. 2013) that depressed people have a smaller online social network. We found the same pattern in the case of cognitive-affective depression. We also found that they posted less in others' timelines, but we did not find that they posted more in their own timeline. Park et al. (2015) found that depressed people posted more in the daytime; we found they posted more in the evening (at least the younger ones). This result is in line with the paper of Joshi and Patwardhan (2020).

However, we have to highlight that it is challenging to compare different studies in the field, as depression and mental health were operationalized in many ways, such as measures from social media. A meta-review of Meier and Reinecke (2021) found a weak correlation between social media usage and depression. They found the effects are complex and vary through depression, mental health indicators, and social media measures.

Earlier studies did not use the Facebook ads interest data. Those classified into the less interest category by Facebook had higher depression levels on both scales. Unfortunately, we do not have a timestamp on the classification, so we cannot follow the interest evolution.

The sub-category results are interesting, but it is not easy to interpret them, and because of the small sample size, the uncertainty is high here. The beer-wine category might indicate frequent going out (with friends), which might be a protective factor. The link between US sports interest and somatic depression is surprising; this interest category is probably a proxy of something different. US sports are broadcasted during the night, so those who have insomnia might watch more US sports broadcasts. Still, this is only a weak assumption. There is mixed evidence in the literature whether dog-cat ownership could reduce depression and loneliness (Gilbey et al. 2007). For older adults, there might be

some protective effects of pet-owning (Branson et al. 2017). However, most of our sample is young or middle age. In these age cohorts, the dog-cat attachment might signify loneliness.

Most of the previous studies on this topic used textual data to identify the markers of depressions. We decided to use only non-textual data in our study. Languages are culturally embedded, and all languages have their own features. It is tough to generalize any results relying solely on textual data. It is challenging to compare the results of the studies relying on different languages. As depression is a global problem, we need to create an analysis framework that could be reproduced easily in different countries. We know that Facebook is a western social media platform, so our analysis has its limits. However, the 2700 million Facebook users of the world are quite a big target group. Our study also has its own culturally dependent factors; the ads interest data is a typical example. The interest in US sports has different meanings in Europe and the US or Asia. As mentioned before, the time difference of US sports coverage might be the mediator variable here and not the US sport that we measured directly. However, the whole framework (using ads interest data to predict depression) could be deployed easily in other countries.

It was also a significant result that depression is not a one-dimension phenomenon. There is quite a considerable debate around the PHQ-9 question module about the number of factors. From a statistical viewpoint, both the one- and two-factor solutions are acceptable. Based on our results, it seems that cognitive-affective and somatic depression have different predictors. This external validation is a strong marker that it is helpful to extract two depression indicators.

The sample size in our study was relatively modest, and our sample was not a probability sample of the Hungarian population. Organic data such as the Facebook data we used here has advantages and disadvantages. These data sources' external reliability is usually higher than survey data, as they measure behavior, not "just" attitudes. Nevertheless, internal reliability (generalizability) is weaker as it is harder to follow strict sampling procedures. The same is true in our study. So, we have to be cautious when we interpret our results. Still, our study confirms some of the previous results, and we did not find any strongly conflicting ones.

The direction of the results is also unambiguous. Smaller networks and lesser activity are the most robust markers of depression. The classification into fewer ads interest categories by Facebook is also a marker of less activity and narrower interests. Nevertheless, different socio-demographic segments have different Facebook usage patterns. So, we have to control our models with demographic variables or use filtered models to achieve reliable results. The sample size of this study made it hard to control the background effects. However, the significant interaction terms presented; sometimes, a social media usage pattern is a marker of depression in one group but not in another group. In our models, age had a more robust interaction effect than gender. Moreover, other variables such as education level or geographical location could also be essential factors. The explanation power of our models was limited. The highest values were around 20 percent. It is a clear sign that there is a space for model improvement. With a bigger sample, it might be possible to test more complex effect directions (not just linear ones). This would be a promising direction for further research. Nevertheless, the generalizability is limited; this study could mark the way for further research.

## 5. Conclusions

Our results have further implications. With the utilization of complex techniques, it is possible to identify the most vulnerable social media users. Models such as ours could help fine-tune the algorithms of social media platforms to support their users with mental problems. However, for this, severe ethical and GDPR concerns have to be considered—Uban et al. (2021) highlighted that if third parties apply these tools, that could harm users' privacy, and they argue for strong ethical statements on how these tools can be used. Graham et al. (2019) summarize some of these ethical challenges. They highlight the problems of biased incoming data, users' low technology literacy, and the distance between

AI developers and clinicians. They suggest that communication about these tools must be transparent and confer that tools are not replacing medical practice, only supplementing them (Graham et al. 2019 We divided the friend number by 100, to help the interpretation of the regression effects. Without this the B coefficients of this variable would be close to zero., p. 16). The uncertainty around this problem is well marked in the OpenAI project (https://openai.com; accessed on 15 January 2022). They are forbidden to use their NLP language models to predict health care problems. This short overview points out that we not only have to develop an accurate algorithm, but we also have to solve ethical and GDPR challenges before using these types of results in real applications.

**Author Contributions:** Conceptualization, Z.K. and K.B.; methodology, Z.K. and K.B.; formal analysis, Z.K.; data curation, Z.K.; writing—original draft preparation, Z.K. and K.B.; writing—review and editing, Z.K.; visualization, Z.K.; funding acquisition, Z.K. All authors have read and agreed to the published version of the manuscript.

**Funding:** The research was funded by the National Research, Development and Innovation Office of Hungary, grant number: FK-128981. The APC was funded by Eötvös Loránd Research Network.

**Institutional Review Board Statement:** Ethical review and approval were waived for this study because data was anonymized within the fieldwork of the study. The results do not allow identification of the individuals involved in the study. The authors and the fieldwork agency managed all information collected in accordance with the General Data Protection Regulation (GDPR).

**Informed Consent Statement:** Informed consent was obtained from all subjects involved in the study.

**Data Availability Statement:** Data is available in the repository of Centre of Social Sciences: https://openarchive.tk.mta.hu/508/.

**Conflicts of Interest:** The authors declare no conflict of interest.

## Appendix A

**Table A1.** Threshold related variation of the average number of close friends and number of people without close friends.

| Threshold (Number of Minimum Reaction per Month | Average Close Friends Number | Number of People without Close Friends |
|---|---|---|
| 1 | 39.59 | 0% |
| 2 | 13.24 | 3% |
| 3 | 6.61 | 8% |
| 4 | 4.02 | 13% |
| 5 | 2.77 | 16% |
| 6 | 2.01 | 22% |
| 7 | 1.55 | 26% |
| 8 | 1.25 | 33% |
| 9 | 1.03 | 38% |
| 10 | 0.86 | 44% |
| 15 | 0.44 | 60% |
| 20 | 0.26 | 69% |
| 30 | 0.11 | 79% |
| 50 | 0.03 | 88% |

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
