# Peer review of "Identifying Depression-Related Behavior on Facebook—An Experimental Study"

_socsci, doi:10.3390/socsci11030135_

Round 1

Reviewer 1 Report

This study proposed a link between depression and the quantity and type of use of Facebook (e.g., those with less friends and posts on their and others’ timelines were more likely to be at-risk of depression). The general Hungarian population and student sample were monitored for their Facebook use over 2 years – the PHQ-9 measured the extent of depression.  

It’s a well-organized and thoroughly presented study although a small sample (it was noted in the conclusion that “it might be possible to test more complex effect directions (not just linear ones)”. There could be better comprehension of the field of research included in the paper. There are a few recent references not included with regards to social media use and behavioral change that have potential in explaining and intervening with mental health (although it is acknowledged that the study focused on depression). It may help to consider how this study fits in with broader studies (other mental health and other communication mediums – not Facebook).

E.g., Meier & Reinecke (2020) found a very small negative effect from the use of social network sites - more comprehensive mental health outcomes were noted as required as well as more rigorous understanding of the characteristics of interactions and transmitted messages (rather than just analyzing screen time).

Meier, A., & Reinecke, L. (2020). Computer-Mediated Communication, Social Media, and Mental Health: A Conceptual and Empirical Meta-Review. Communication Research, 48(8), 1182–1209. doi:10.1177/0093650220958224

E.g., Joshi & Patwardhan (2020) used an unsupervised approach (clustering via machine learning algorithms that used NLP) in an analysis to understand users’ behavioral features with a social network site (mainly Twitter) and distinguished normal users from at-risk users (the latter were characterized by the scale of change of their use).

Joshi, D., & Patwardhan, D. M. (2020). An analysis of mental health of social media users using unsupervised approach. Computers in Human Behavior Reports, 2, 100036. doi:10.1016/j.chbr.2020.100036

E.g., Uban et al., (2021) is a similar study to Joshi & Patwardhan (2020) – these authors identified patterns of language in social media users aimed to distinguish between users diagnosed with a mental disorder and healthy users with a model of emotion evolution to assist clinicians in diagnosing patients (with depression, anorexia, and self-harm tendencies).

Uban, A.-S., Chulvi, B., & Rosso, P. (2021). An emotion and cognitive based analysis of mental health disorders from social media data. Future Generation Computer Systems, 124, 480–494. doi:10.1016/j.future.2021.05.032

“Our results have further implications. With the utilization of complex techniques, it is possible to identify the most vulnerable social media users. Models like ours could help fine-tune the algorithms of social media platforms to support their users with mental problems”.

It’s a piece of the puzzle but has not been presented in such a way. E.g., A review by Graham et al. (2019) of AI and mental health outlined data sources as electronic health records, mood rating scales, brain imaging data, monitoring systems and social media platforms to predict, organize, or subgroup a range of mental ill-health and suicidality.

Graham, S., Depp, C., Lee, E. E., Nebeker, C., Tu, X., Kim, H.-C., & Jeste, D. V. (2019). Artificial Intelligence for Mental Health and Mental Illnesses: an Overview. Current Psychiatry Reports, 21(11). doi:10.1007/s11920-019-1094-0

Author Response

We want to thank the reviewer for the suggested literature. We built them into the paper. This research area is developing rapidly, and these new papers all fit well to our introduction and discussion. We are particularly grateful for the Graham et al. paper, as our suggestions about how these results could be used further were not too exact in the previous version of the paper.

Reviewer 2 Report

The article should give more importance to the final usefulness of using social networks as a detector of mental illness or emotional discomfort. The sample is small to implement big data algorithms, the authors already explain it, but the small sample means that some conclusions have very subjective explanations. Perhaps those subjective opinions, such as the one that refers to American sports, should be suppressed. Regarding the possible uses of these detection systems, it would be necessary to review the literature on the conflicts between detection and privacy rights.

Author Response

Thank you for reviewing our paper. The reviewer is correct; we have some speculative results, especially about the American football role. We deleted that part and added a sentence about the uncertainty of the results because of the small sample size. We also extended the part about how these detection algorithms can be used and what are the most vital ethical and privacy challenges.

Reviewer 3 Report

Greater clarity is needed on analysis.

Author Response

Thanks for the detailed comments; they were helpful in improving our paper. We made a language editing; We hope the paper is smoother now. Our detailed answers are available in the uploaded pdf.

Round 2

Reviewer 3 Report

Thank you for addressing my concerns.  The explanations added were helpful to understanding the analysis.  There remains some English editing that is needed on lines 28, 316, 326 and 338.  In general, the methods and results should be written in the past tense and there are places that are in the present or future tense.

Author Response

Thank you for the language correction; we corrected the highlighted problems, and some others too, where we used the present tense, not past, in the data and results part.